# Contractile Behavior of Right Atrial Myocardium of Healthy Rats and Rats with the Experimental Model of Pulmonary Hypertension

**DOI:** 10.3390/ijms23084186

**Published:** 2022-04-10

**Authors:** Oleg Lookin, Elena Mukhlynina, Yuri Protsenko

**Affiliations:** Institute of Immunology and Physiology of Ural Branch of Russian Academy of Sciences, 106 Pervomayskaya St., 620049 Yekaterinburg, Russia; elena.mukhlynina@yandex.ru (E.M.); y.protsenko@iip.uran.ru (Y.P.)

**Keywords:** rat atrial myocardium, pulmonary hypertension, heart failure, isometric force–length relationship, slow force response, post-rest potentiation

## Abstract

There is a lack of data about the contractile behavior of the right atrial myocardium in chronic pulmonary heart disease. We thoroughly characterized the contractility and Ca transient of isolated right atrial strips of healthy rats (CONT) and rats with the experimental model of monocrotaline-induced pulmonary hypertension (MCT) in steady state at different preloads (isometric force-length), during slow force response to stretch (SFR), and during post-rest potentiation after a period of absence of electrical stimulation (PRP). The preload-dependent changes in the isometric twitch and Ca transient did not differ between CONT and MCT rats while the kinetics of the twitch and Ca transient were noticeably slowed down in the MCT rats. The magnitude of SFR was significantly elevated in the MCT right atrial strips and this was accompanied by the significantly higher elevation of the Ca transient relative amplitude at the end of SFR. The slow changes in the contractility and Ca transient in the PRP protocol did not differ between CONT and MCT. In conclusion, the alterations in the contractility and Ca transient of the right atrial myocardium of monocrotaline-treated rats with pulmonary hypertension mostly concern the elevation in SFR. We hypothesize that this positive inotropic effect in the atrial myocardium may (partly) compensate the systolic deficiency of the right ventricular failing myocardium.

## 1. Introduction

Pulmonary heart disease is a severe pathological state accompanied by systolic/diastolic dysfunction, predominantly of the right heart. The triggering mechanism of the pathogenesis of pulmonary heart disease is a progressive increase in the resistance of the pulmonary vessels, which causes pathological remodeling of the right ventricle (RV) via numerous pathways [1,2,3,4]. In the normal heart, the systolic efficiency of the RV is not solely determined by its contractility but is also modulated by a rapid extra pre-filling of RV by the contracting right atrium (RA). This RA-induced additional ejection of a small volume of blood into the RV elevates its filling volume by ~20–30% and thus substantially contributes to the systolic ejection [5,6,7,8]. It remains unclear whether and how this physiological mechanism can be affected by pulmonary heart disease. To answer this issue, it is needed to characterize the contractile function of the right atrial myocardium both in normal conditions and in right heart failure.

There are few reports about the volume-related effects on the contractility of the whole atrium (i.e., the Frank–Starling mechanism, FSM) under in vivo conditions [9,10,11] and much less has been studied on the level of isolated muscle strips: the level where more specific biomechanical interventions could be studied to characterize the contractility of the atrial myocardium in detail. On atrial strips from human healthy donor hearts, it has been shown that length-dependent changes do exist and they relate not only to the mechanical characteristics of isometric twitch (FSM) but also to the kinetic characteristics of cytosolic calcium [12]. Compared to the ventricular myocardium, a weaker length-dependent modulation of atrial myocardial contractility has been shown in different mammals [13]. The contemporary changes in the mechanical activity and Ca transient in the atrial myocardium have been analyzed in a few studies [12,14], but these studies are limited to mechanically unloaded conditions or, at best, to different preloads.

The FSM is the mechanism of rapid modulation of contractility due to the preload-induced increase in Ca^2+^ demand by myofilaments and it has been shown that this mechanism is impaired in heart failure [15,16]. In addition, there is also a medium-term adaptation mechanism to rapid change in preload: the slow force response (SFR) or the Anrep effect for the whole heart. After the rapid stretch of the myocardium, this mechanism provides a slow increase in the isometric tension by 20–30% relative to the value immediately after stretch [17]. The SFR is differently expressed in the ventricular and atrial myocardium while very limited experimental data exist in the latter case [18,19]. Our previous studies have demonstrated that the SFR is suppressed in RV trabeculae of failing rat myocardium and that this is accompanied by changes in the kinetics of Ca^2+^ with the regulatory protein troponin C [20,21]. At present, there is no data on how the SFR is altered in the failing atrial myocardium, except a study on human end-stage failing myocardium [19]. In addition, no evidence exists on the alterations in Ca^2+^ regulation in the healthy and failing RA myocardium. For example, it remains unknown if the kinetics of the Ca transient and SR capacity for Ca^2+^ are altered in heart failure.

In the present study, we thoroughly characterized the contractile behavior and Ca transient in isolated right atrial strips from healthy rats and rats with a monocrotaline model of pulmonary heart disease in order to reveal functional remodeling of length-dependent regulation in failing atrial myocardium. We compared the characteristics of simultaneously measured isometric twitches and Ca transients in three distinct biomechanical tests: force-length (the short-term regulation of contractility), slow force response (the middle-term regulation), and post-rest potentiation (the latter allows for indirect assessment of the Ca^2+^ capacity of the SR). Our findings indicate that the preload-dependent and Ca^2+^-related regulation of contractility in rat atrial myocardium is altered under monocrotaline-induced pulmonary heart disease.

## 2. Results

### 2.1. Morphometric and Histological Analysis of Atrial Myocardium

The morphometric indices for the heart chambers of CONT and MCT rats are shown in Table 1. The average body weight of MCT rats was significantly reduced relative to the CONT rats. This was accompanied by a significant increase in the average heart weight (both ventricles plus septum) and right ventricle weight (but not the left ventricle) of the MCT rats. While the average tibia length was no different between CONT and MCT rats, the ratio of the right ventricle weight to the tibia length was significantly larger in the MCT rats, indicating severe right ventricular hypertrophy in these rats.

The pulmonary trunk wall of MCT rats had significant thickening of the tunica media, with degenerative changes in smooth muscle cells, edema and foci of destruction, and endothelial desquamation (Figure 1). These histopathological changes can be considered as pathognomonic histological features of pulmonary arterial hypertension. The histopathological changes were also characterized by the development of myocarditis.

Moderate diffuse inflammatory cell infiltration and focal loss of cross-striations were detected in the right atrium of MCT rats (Figure 2). Nuclear polymorphism was moderate. The observed atrial epicardium thickening was accompanied by perivascular and interstitial edema formation.

The morphometric analysis revealed hypertrophy of the cardiomyocytes. An increase in the average myocyte diameter and right atrial wall thickness was observed in the MCT rats (Table 2). In contrast, the number of cardiomyocyte nuclei per mm^2^ of a cardiac section was decreased in these rats. The morphometric changes are indicative of pathological remodeling in the right ventricular and right atrial myocardium of the MCT rats.

### 2.2. Preload-Dependent Effects in Atrial Myocardium

The effect of preload on the kinetics of force/Ca transient development and the appropriate differences between the atrial strips of CONT and MCT rats are shown in Figure 3. To demonstrate it clearly, we used self-normalized traces of isometric twitches and Ca transients, simultaneously measured during the force-length (F-L) protocol. While no significant differences were found in the amplitude of the isometric twitch (see Figure 4), the duration of the twitch and Ca transient was noticeably longer in the MCT strips within the entire preload range. In both the CONT and MCT strips, preload had a moderate effect on the kinetics of force development and decline while almost no such effect was observed in regard to the Ca transient decay characteristics (Figure 3).

The isometric tension–length relationships, obtained for both peak active and passive tension values, did not differ between CONT and MCT atrial strips (Figure 4A,B). In both groups, the averaged passive tension under the highest preloads (95% and 100% *L_MAX_*) was even larger than the peak active tension under the same preloads. The maximal rate of isometric tension rise, if normalized to the peak active isometric tension, was significantly lower in MCT vs. CONT group under any preload (Figure 4C, *p* < 0.05). In contrast, the maximal normalized rate of isometric relaxation did not differ between CONT and MCT (Figure 4D). The time-to-peak of isometric tension was significantly higher in the MCT atrial muscles compared to the CONT group within the entire preload range (Figure 4E). On average, for the preload range of 75–100% *L_MAX_*, this value was 21.5 ± 1.5% higher in the MCT atrial strips. In contrast, the relaxation time of isometric tension from the peak value to 50% of its amplitude was not significantly different between the CONT and MCT strips under any preload (Figure 4F).

The preload-dependent changes in the diastolic level or amplitude of Ca transient did not differ between the CONT and MCT groups (Figure 5A,B). However, the rate of Ca transient rise, if evaluated as normalized to the Ca transient amplitude, was significantly lower in the MCT atrial strips compared to the CONT strips by ~20% on average (Figure 5C, *p* < 0.05 for any preload). Consequently, the time-to-peak of Ca transient was significantly higher by 42.6 ± 1.3% on average in the MCT group under all preloads (Figure 5E). Similarly, the maximal normalized rate of Ca transient decay was significantly lower (except the lowest preload of 75% *L_MAX_*) and therefore the time of decay from the peak to 50% amplitude, *T*_50_, was significantly longer (under all preloads) in MCT vs. CONT atrial strips (*p* < 0.05, Figure 5D,F, respectively). In the MCT atrial strips, the relative prolongation of the Ca transient decay, which was characterized by the decay time from the peak to 50% amplitude, was approximately 20% under all preloads. This value was smaller compared to the relative prolongation of the Ca transient rising phase (as characterized by time-to-peak, by ~43%). However, the absolute prolongations of time-to-peak and *T*_50_ were nearly similar and, if averaged for all the preloads, these values amounted to 14.2 ± 0.4 and 9.1 ± 1.3 ms, respectively.

### 2.3. Slow Force Response in Atrial Strips

The representative slow force responses (SFRs) measured in the atrial strips of CONT and MCT rats after rapid stretch from 85% to 95% *L_MAX_* are shown in Figure 6A,D, respectively. In addition, the corresponding individual traces of the isometric twitch and Ca transient measured at some time points during the SFR (Figure 6B,C,E,F) demonstrate that the significant changes in the characteristics of the force/Ca transient occurred immediately after the stretch of the strip. However, during the further development of SFR, both isometric twitch and Ca transient displayed very minor changes in their kinetics.

The slow changes in the characteristics of the individual isometric force twitch or Ca transient, as it can be seen during SFR, are shown in Figure 7. The magnitude of SFR, i.e., the percent ratio of peak active force at the end of SFR to the value for the first twitch after stretch, was significantly higher in the MCT vs. CONT atrial strips: 141.0 ± 22.9% vs. 116.8 ± 12.7%, *p* = 0.013 (Figure 7A). This difference was in concordance with the relative change in the Ca transient amplitude at the end of SFR. This change was 104.3 ± 14.6% in the MCT atrial strips and 90.6 ± 10.8% in the CONT atrial strips, respectively (*p* = 0.023, Figure 7E). Moreover, the elevated SFR in the MCT atrial strips was accompanied by an increased relative elevation in the time of relaxation from the peak force to 50% of its amplitude, but this relative increase was not significantly different between the CONT and MCT groups (Figure 7C). In both groups, the relaxation time and time-to-peak force were rapidly elevated immediately after stretch and then were virtually unchanged during the further progression of the SFR (Figure 7B,C, respectively). The time of the Ca transient decay from the peak to 50% of its amplitude was found to be insensitive to the stretch and did not display any slow modifications during the SFR (Figure 7F). Finally, the CONT and MCT atrial strips showed a slow increase in the Ca transient diastolic level over SFR. This elevation was as small as 1% of the value immediately after stretch (Figure 7D). However, in the combination with small changes in the Ca transient amplitude, these minor changes in the diastolic Ca^2+^ level may substantiate the significant effect on the peak isometric force. 

### 2.4. Post-Rest Potentiation in Atrial Strips

To assess the capacity of the sarcoplasmic reticulum in atrial strips of healthy and monocrotaline-treated rats, we implemented the post-rest potentiation protocol, where a strip was placed under rest without electrical pacing for 60 s. Immediately after the resumed stimulation, the atrial strip developed first twitches that were higher compared to the twitch immediately before the pause (Figure 8A). The extent of post-rest potentiation of active isometric tension was 297 ± 85% and 269 ± 102% in the CONT and MCT atrial strips, respectively (no significant difference, Figure 8B). The kinetics of the slow change in the twitch amplitude after resumption of the stimulation also did not differ between these groups. Similarly, the Ca transient diastolic level and amplitude showed that the post-rest extent and kinetics did not differ between CONT and MCT (Figure 8C,D). In both groups, during the first several twitches after the resumption of the stimulation, the diastolic level of Ca transient was slightly decreased but then further elevated above the level before the pause (Figure 8C). In contrast, the Ca transient amplitude in the first post-pause twitch was 118.7 ± 10.9% and 109.5 ± 12.7% of the value in the twitch before the pause in CONT and MCT, respectively (Figure 8D, no significant difference). 

## 3. Discussion

In this study, we thoroughly characterized the contractile behavior and intracellular Ca^2+^ transient measured in isometrically contracting isolated strips of the right atrial myocardium of healthy rats and rats with the experimental model of monocrotaline-induced pulmonary hypertension followed by right heart remodeling and end-stage failure. The preload-dependent changes in the isometric peak active and passive tension in rat atrial strips were found to be almost unaffected by the development of monorotaline-induced pulmonary hypertension. This result indicates that the Frank–Starling mechanism (FSM) remains unaltered in the atrial myocardium of monocrotaline-treated rats. Previously published studies showed that FSM is noticeable in the whole atrium under in vivo conditions [9,10,11] and in isolated atrial muscles [12]. The preload-dependent changes in the active isometric tension of human atrial myocardial strips are accompanied by changes in the systolic and diastolic levels of cytosolic calcium [12], although in our study we did not show such length dependence. 

In the present study, we showed for the first time that the isolated right atrial strips of healthy and monocrotaline-treated rats developed a noticeable slow force response (SFR) to rapid stretch. We previously showed suppression of SFR in the right ventricular trabeculae of MCT rats and established the role of Ca–TnC interaction in this phenomenon [20]. To date, it is known that SFR is manifested at all structural levels of myocardial tissue: from a single cardiomyocyte to a whole chamber [17,18,21,22,23,24,25] and it provides ~20–30% additional contractility relative to the state immediately after stretch. The exact mechanism(s) of extra calcium that provides the development of SFR remains unknown [17,26]. The physiological role of SFR is a medium-term adaptation (i.e., during minutes) of myocardial contractility to changes in the pre- and/or afterload. The additional pre-loading of the ventricle by atrial contraction and small-volume extra ejection from the atrium into the ventricle before ventricular systole has been demonstrated in a whole heart study [27]. This study showed that if sequential activation of the atrium and ventricle is turned on (the correct pattern of activation), the pressure in the ventricle begins to change slowly and significantly while under synchronous activation of the atrium and ventricle, this slow pressure response in the ventricle is inverted and its extent is decreased. This change matches the character and extent of SFR that characterizes the ventricular myocardium of many warm-blooded and human species [17,19,20,23,25,28]. It is hypothesized that the above-described maladaptive changes in the SFR of right ventricular myocardium can be compensated (at least in part) by an elevated atrial SFR that provides adequate pre-distension of the ventricular chamber and ensures sufficient ejection [7,29]. However, further elucidation is needed to clarify this atrial-ventricular functional interaction.

We found that isolated right atrial strips from control and monocrotaline-treated rats accumulated similar relative amounts of extra Ca^2+^ during the rest period, during which no electrical pacing was applied (post-rest potentiation, [30]). This indicates that the Ca^2+^-handling pathways in rat right atrial myocardium remain unaffected by the development of pulmonary hypertension. It has been shown that the post-rest potentiation decreases in cardiac pathologies, e.g., in dilated cardiomyopathy [31]. However, these studies were carried out on ventricular myocardium and there is a lack of reports on the Ca^2+^ regulation in atrial myocardium. This motivates us to address this issue in further investigations.

Some of our findings about the kinetics of Ca transient in rat right atrial strips do not agree with previously published data. It has been shown on human myocardial tissue [32,33,34] that in the atria, the rapid phase of elevation of cytosolic Ca^2+^ occurs much faster and the time to reach its peak is much earlier than in the ventricular myocardium. However, human and rat myocardium differs greatly in the mechanisms of electromechanical coupling, so these contradictions may have a real basis, and may not be caused by differences in methodological approaches and experimental conditions.

The above-described results obtained in rat atrial myocardium can be compared with already known mechanical and calcium-related characteristics of the ventricular myocardium. The following differences in the characteristics of the contractility and calcium kinetics between right atrial and right ventricular myocardium of rats can be highlighted: (i) the atrial myocardium develops lower active isometric tension (FSM is less effective); (ii) the mechanical tension develops and declines faster in the atrial myocardium; (iii) the maximal rate of cytosolic Ca^2+^ elevation is lower and the time to peak of Ca transient is longer in the atrial myocardium; and (iv) the Ca transient decline does not show a “bump” phase, which is typical in the ventricular myocardium [21,24]. Compared to the ventricular myocardium, the length-dependent modulation of atrial myocardial contractility is much weaker in many mammals [13]. In addition, in rats with monocrotaline-induced right heart remodeling, SFR is increased in the right atrial myocardium (the present results) while it is decreased in the right ventricular myocardium, as we previously reported [20]. This is in agreement with previously published data indicating that human failing myocardium can develop significant SFR [35]. However, our results contradict the findings that human non-failing myocardium produces higher SFR compared to failing human myocardium [28]. The structural and functional features of the mechanisms of electromechanical coupling in atrial and ventricular cardiomyocytes may explain these differences [34,36,37], and the differential alterations under heart failure [38,39]. Nevertheless, the mechanical response and Ca transient are prominently slowed down in the failing right atrial muscles and this conforms very well with the previously established deceleration of the contractile response in the right ventricular myocardium [1,22,40]. It may be concluded in this context that the myocardium of atria and ventricles undergoes similar pathological changes in the regulation of contraction. 

In conclusion, the right atrial myocardium of rats with monocrotaline-induced pulmonary hypertension followed by right heart remodeling and failure displays minor shifts in its contractile behavior and Ca^2+^ regulation relative to healthy rat atrial myocardium. The observed alterations mostly concern the slow force response, which is elevated in the failing heart. The evaluation in the contractile function of atrial myocardium in right heart failure has clinical significance because the right ventricular systolic function in the failing heart can be further impaired under the increased pressure that develops in the right atrium [41].

## 4. Materials and Methods

### 4.1. Ethical Approval

The animals involved in the present study were cared for according to the Directive 2010/63/EU of the European Parliament and the Guide for the Care and Use of Laboratory Animals published by the US National Institutes of Health (NIH Publication No. 85-23, revised 1985), and their use was approved by the local Institutional Ethics Committee. Male Wistar rats, aged 1.5–2 months, were obtained from the Institutional Animal House and maintained under standard conditions (12h light/dark cycle, water and food ad libitum). The animals were rapidly euthanized by cervical dislocation immediately before tissue collection. The animals were injected intramuscularly with heparin sodium (5000 IU/kg) to prevent blood clotting in coronary vessels, anesthetized 30 min later with Zoletil-100 (0.3 mL/kg body weight) and 2% Xylazine (1 mL/kg body weight), and rapidly euthanized 15–20 min later by cervical dislocation immediately before the removal of the heart.

### 4.2. Monocrotaline-Induced Experimental Model of Pulmonary Hypertension

The male rats were either healthy (CONT, *n* = 10) or treated with monocrotaline to induce pulmonary hypertension followed by right ventricular hypertrophy and heart failure (MCT, *n* = 10) as described in detail elsewhere [22]. Briefly, 4-week-old rats were given a single injection of saline with pyrrolizidine alkaloid monocrotaline (2 mL/kg; final concentration 50 mg/kg body weight). The CONT rats were injected with an equivalent volume of saline solution. MCT rats were euthanized as soon as the loss of body weight and dyspnea at rest became noticeable (~3.5–4 weeks after the treatment) and CONT rats were euthanized at the same age (~7.5–8 weeks). 

### 4.3. Histological Analysis of Atrial Myocardium and Pulmonary Trunk

Initially, the hearts with pulmonary trunks were fixed in 4% paraformaldehyde for 48 h. The fixed right atrium and pulmonary trunks were processed, embedded in paraffin blocks, and cut into 3–5-μm-thick sections. The slides were stained with hematoxylin and eosin (H&E, Biovitrum, Saint-Petersburg, Russia) and Picrosirius Red (Picro Sirius Red Stain Kit (ab150681), Abcam, Cambridge, UK). The sections were examined under the light microscope Leica DM2500 (Leica, Wetzlar, Germany). The atrial wall thickness, cardiomyocyte diameters, and number of cardiomyocyte nuclei per mm^2^ were assessed using the image analysis software Leica Application Suite 4.9 (Leica, Germany) and VideoTest “Morphology” 5.2 (VideoTesT, Saint-Petersburg, Russia).

### 4.4. Functional Measurements in Isolated Atrial Muscle Strips

The heart was removed immediately following euthanasia and placed in a modified Krebs-Henseleit solution (KHB, in mM: NaCl 118, KCl 4.7, MgSO_4_ 1.2, KH_2_PO_4_ 1.2, NaHCO_3_ 25, HEPES 10, CaCl_2_ 2, glucose 11.1, pH adjusted to 7.35 at 25 °C under aeration with 95% O_2_/5% CO_2_), containing 30 mM 2,3-butanedione monoxime (BDM). A thin right atrial strip (150–300 μm thick, 1–3 mm long) was dissected from the heart and attached to a force transducer and length servomotor in the experimental chamber under continuous perfusion by BDM-free KHB (saturated by 95%O_2_/5%CO_2_). To measure Ca transients, the strip was preincubated in saline with 5 μM fura-2/AM + 0.4% *w*/*v* Pluronic F-127. Atrial strips (*n* = 12 for CONT/MCT) were tested in 3 runs: (1) isometric force-length (F-L) protocol, (2) slow force response (SFR) protocol, and (3) post-rest potentiation (PRP) protocol. All measurements were conducted in muscle preparations under electrical impulses at 2 Hz and 30 °C. 

To implement the F-L protocol, a muscle strip was fully released to produce zero passive tension and then gradually stretched in 50 μm steps until preload *L_MAX_* (maximal active tension) and allowed to equilibrate under each new preload prior to the measurement of its active/passive tension and Ca transient under this preload. The measurements were done under the following preloads: 75% (minimal active tension), 80%, 85%, 90%, 95%, and 100% *L_MAX_*. For each preload, ~30 consecutive force/Ca transient twitches were averaged to improve the signal-to-noise ratio. The series of isometric contractions and Ca transients, as they were obtained under different preloads under steady-state contraction, were used to construct isometric force–length (tension–length) relationships and corresponding preload-dependent relationships for various characteristics (amplitude, rate, timing) of the isometric force and Ca transient traces. The tension values were obtained from force values by normalizing them to the cross-section of the atrial strip; a cross-section was calculated assuming the following elliptic formula: S = πd^2^/12, where d is the larger width of the strip measured in the transversal direction.

After the completion of the F-L protocol, the length of a muscle strip was set to 85% *L_MAX_*. After equilibration for ~10 min, the strip was then subjected to rapid (during 100 ms) stretch to 95% *L_MAX_* and slow twitch-by-twitch changes in contractility and Ca-transient were simultaneously recorded (SFR protocol). The typical duration of the slow changes for our experimental conditions (pacing at 2 Hz and 30 °C) was approximately 3 min. The slow changes in the characteristics of isometric tension/Ca transient in an atrial strip were evaluated as a % of corresponding characteristics measured in the first twitch immediately after stretch.

Finally, the atrial strips from the CONT and MCT groups were tested for their ability to accumulate Ca^2+^ during the long period of rest, during which no electrical pacing was applied (PRP protocol). The tests were carried out under 2 preloads: 85 and 95% *L_MAX_*; the duration of pause was 60 s. The mechanical state and fluorescent levels in the atrial strip were measured before, during, and after the rest. The maximal effect of the rest period on the force/Ca transient amplitude was evaluated as a % of that value before the rest period.

All chemicals were purchased from Sigma-Aldrich (St Louis, MO, USA).

### 4.5. Data Acquisition

The isometric contraction, actual length of a muscle strip, and fura-2 fluorescence were simultaneously measured using the Muscle Research System (Scientific Instruments, Heidelberg, Germany) and sampled at 10 kHz via DAC/ADC (PCI-1716S; AdLink Technology, New Taipei City, Taiwan) using custom-made software, which was run in a real-time environment (HyperKernel; Nematron, Ann Arbor, MI, USA) integrated into Windows XP (Microsoft; Redmond, WA, USA). Ca transients were obtained as a ratio of the fluorescence signals obtained at 510 nm by excitation at 340/380 nm (*F* = *F*_340_/*F*_380_) and presented here as *F*/*F*_0_, where *F*_0_ is the ratio measured in the quiescent non-stretched atrial strip. 

### 4.6. Statistical Analysis

The Mann–Whitney U test was used to evaluate the significance of the difference in the morphometric data between CONT and MCT rats. Kruskal–Wallis ANOVA was used to evaluate the significance of the difference between the mean values of the twitch/Ca transient characteristics (amplitude, time-to-peak, time to half-relaxation) for CONT and MCT strips under a fixed preload. The differences were considered significant at *p* < 0.05. Data are presented as mean ± S.D.

## Figures and Tables

**Figure 1 ijms-23-04186-f001:**
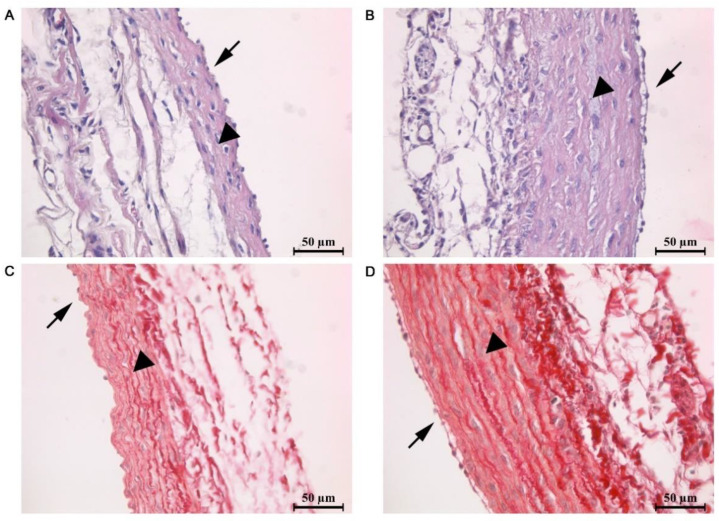
Representative microphotographs of pulmonary trunk cross-sections in control rats (**A**,**C**) and rats with the experimental model of pulmonary hypertension (**B**,**D**). (**A**,**C**) Staining by hematoxylin and eosin (H&E). (**B**,**D**) Staining by Picrosirius Red. All images were obtained with 40× magnification. Arrowheads indicate the increased tunica media thickness in the MCT rats; arrows show tunica intima with endothelial cells.

**Figure 2 ijms-23-04186-f002:**
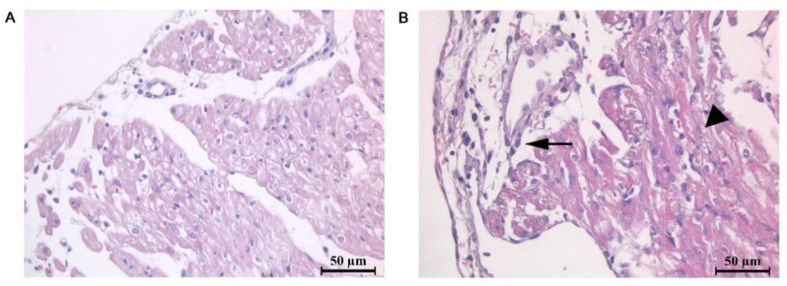
Representative microphotographs of the right atrium frontal section in control rats (**A**) and rats with the experimental model of pulmonary hypertension (**B**). Staining by hematoxylin and eosin (H&E). All images were obtained with 40× magnification. Arrow indicates epicardium thickening, inflammatory cell infiltrates, and interstitial and perivascular edema, and the arrowhead points to the hypertrophic and damaged atrial cardiomyocytes.

**Figure 3 ijms-23-04186-f003:**
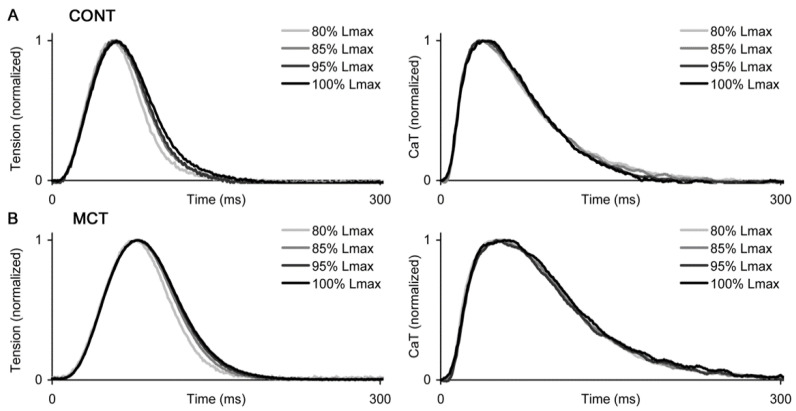
The representative self-normalized traces of isometric force (left panels) and Ca transients (CaT, right panels) measured in the right atrial strips under different preloads (within a force-length protocol, indicated as a % of *L_MAX_*). (**A**) Atrial strips of control rats (CONT). (**B**) Atrial strips of monocrotaline-treated rats (MCT). Each individual trace is normalized to its peak. The measurements were done at 30 °C and a 2 Hz pacing rate.

**Figure 4 ijms-23-04186-f004:**
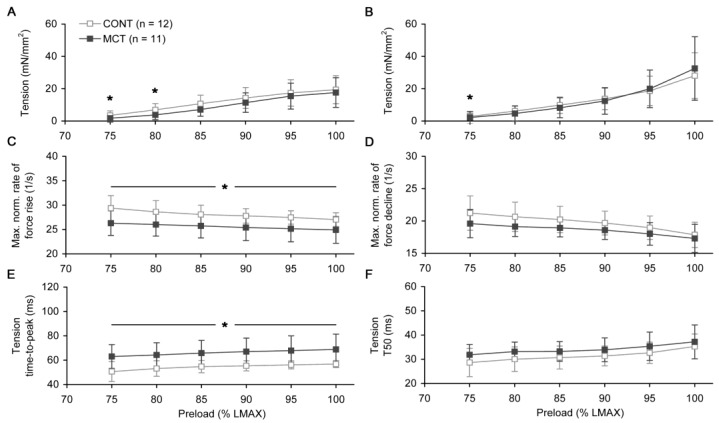
The preload-dependent changes in the mechanical characteristics of steady-state isometric twitch measured in the right atrial strips of control (CONT, *n* = 12) and monocrotaline-treated (MCT, *n* = 11) rats. (**A**) Tension–length relationship for peak active tension. (**B**) Tension–length relationship for passive tension. (**C**) Maximal normalized rate of force rise. (**D**) Maximal normalized rate of force decline. (**E**) Time-to-peak of isometric tension. (**F**) Relaxation time of isometric tension from the peak to 50% of its amplitude (*T*_50_). The legend in (**A**) is common for all panels. Data shown as mean ± S.D. *—the difference between CONT and MCT is significant at *p* < 0.05. All measurements were done at 30 °C and a 2 Hz pacing rate.

**Figure 5 ijms-23-04186-f005:**
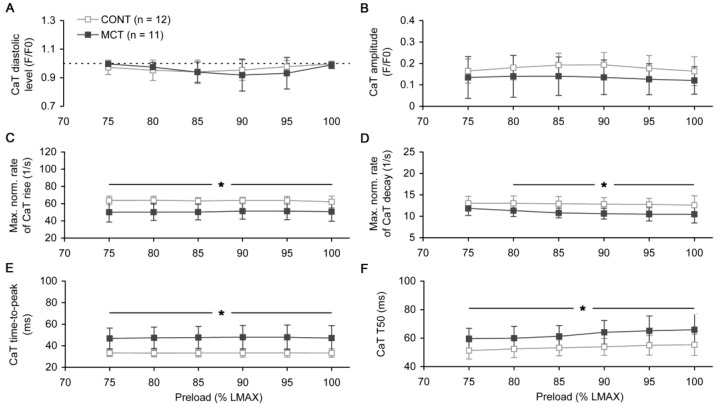
The preload-dependent changes in the characteristics of Ca transient (CaT) measured in the steady-state isometric twitch of the right atrial strips of control (CONT, *n* = 12) and monocrotaline-treated (MCT, *n* = 11) rats. (**A**) Diastolic level of CaT. (**B**) Amplitude of CaT. (**C**) Maximal normalized rate of CaT rise. (**D**) Maximal normalized rate of CaT decay. (**E**) Time-to-peak of CaT. (**F**) Decay time of CaT from peak to 50% of its amplitude (*T*_50_). The legend in panel (**A**) is common for all panels. Data in panels (**A**,**B**) are shown in *F*/*F*_0_ units, where *F*_0_ is the intensity of fura-2 measured in a quiescent atrial preparation at slack length. Data shown as mean ± S.D. *—the difference between CONT and MCT is significant at *p* < 0.05. All measurements were done at 30 °C and a 2 Hz pacing rate.

**Figure 6 ijms-23-04186-f006:**
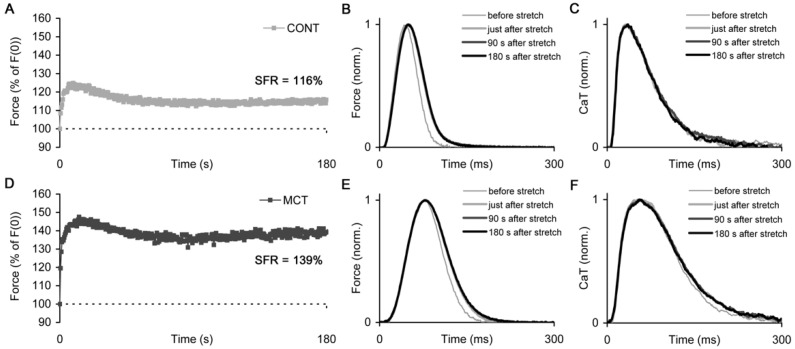
The representative slow force response (SFR) measured in the right atrial strip of control (CONT) and monocrotaline-treated (MCT) rats. (**A**) The SFR in an exemplary CONT atrial strip. (**B**,**C**) The individual traces of the isometric force and Ca transient measured during the development of SFR of the exemplary CONT atrial strip. (**D**) The SFR in an exemplary MCT atrial strip. (**B**,**C**) The individual traces of the isometric force and Ca transient measured during the development of SFR of the exemplary MCT atrial strip. The magnitude of SFR for these exemplary atrial strips is shown in panels (**A**,**D**). The time points for the curves in panels (**B**,**C**,**E**,**F**) are shown in the appropriate legends. The force and Ca transient traces in panels (**B**,**C**,**E**,**F**) are normalized to their respective peaks.

**Figure 7 ijms-23-04186-f007:**
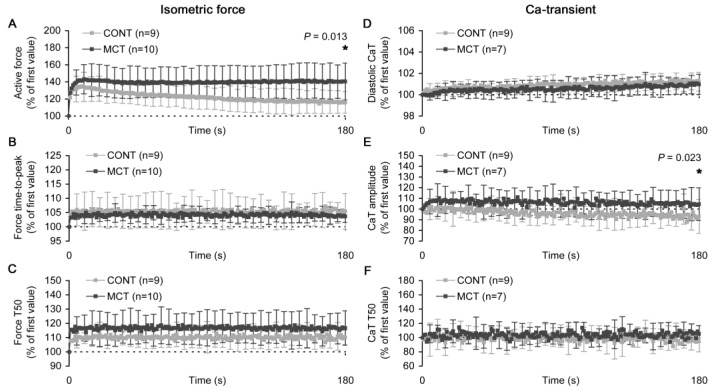
The slow changes in the characteristics of individual isometric force twitches (left panels) and Ca transient (CaT, right panels) measured in the right atrial strip of control (CONT) and monocrotaline-treated (MCT) rats after rapid stretch (during 100 ms) of a strip from 85% to 95% *L_MAX_*. (**A**) The amplitude of active force. (**B**) The time-to-peak of active force. (**C**) The time of relaxation from peak force to 50% of its amplitude (*T*_50_). (**D**) Ca transient diastolic level. (**E**) Ca transient amplitude. (**F**) The time of Ca transient decay from peak to 50% of its amplitude (*T*_50_). All values are shown as a % of the value in the first twitch after stretch; dotted horizontal lines represent a 100% value (no change relatively to the first twitch). The number of atrial strips is indicated in the legends. Data shown as mean ± S.D. *—the difference between CONT and MCT at the end of the slow changes is significant at *p* < 0.05. All measurements were done at 30 °C and a 2 Hz pacing rate.

**Figure 8 ijms-23-04186-f008:**
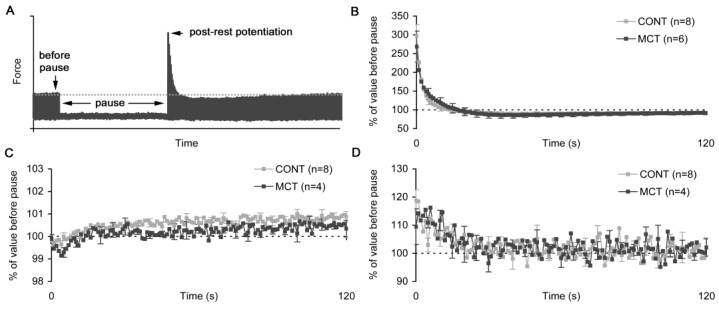
The effect of a pause in the stimulation (resting period) on the post-rest potentiation in the isometric twitch and Ca transient diastolic level and amplitude in the right atrial strip of control (CONT) and monocrotaline-treated (MCT) rats. (**A**) Representative plot of the post-rest potentiation of isometric twitches after a pause in the stimulation. Dotted lines show the “no effect” level. (**B**) The amplitude of the active force. (**C**) Ca transient diastolic level. (**D**) Ca transient amplitude. All values are shown as a % of the value measured before pause. The number of atrial strips is indicated in the legends. Data shown as mean ± S.D. All measurements were done for the preload of 95% *L_MAX_* at 30 °C and a 2 Hz pacing rate.

**Table 1 ijms-23-04186-t001:** Morphometric indices for healthy (CONT) and monocrotaline-treated (MCT) rats.

	CONT (*n* = 10)	MCT (*n* = 10)
Body weight (BW, g)	184 ± 38	165 ± 39 *
Heart weight (HW, g)	0.51 ± 0.06	0.59 ± 0.09 *
HW/BW ratio (×10^3^)	2.73 ± 0.26	3.68 ± 0.55 *
Left ventricle weight (LVW, g)	0.24 ± 0.03	0.20 ± 0.05
Right ventricle weight (RVW, g)	0.11 ± 0.01	0.23 ± 0.05 *
Tibia length (TL, mm)	33.2 ± 1.9	33.5 ± 2.5
LVW/TL ratio (×10^3^)	7.2 ± 0.8	6.1 ± 1.1 *
RVW/TL ratio (×10^3^)	3.3 ± 0.3	6.8 ± 1.3 *

Data shown as mean ± S.D. *—significant difference between CONT and MCT rats (*p* < 0.05, Mann–Whitney U test).

**Table 2 ijms-23-04186-t002:** Morphometric features of right ventricular (RV) and right atrial (RA) tissue of control rats (CONT) and rats with the experimental model of pulmonary hypertension (MCT).

	CONT (*n* = 6)	MCT (*n* = 5)
Diameter of RV cardiomyocytes, µm	14.0 ± 2.3	16.3 ± 2.2
Diameter of RA cardiomyocytes, µm	8.3 ± 0.8	10.5 ± 0.3 *
Thickness of RV wall, µm	626 ± 86	821 ± 64 *
Thickness of RA wall, µm	11.8 ± 2.5	38.7 ± 13.2 *
Number of nuclei in RV cardiomyocytes per mm^2^	1628 ± 394	1213 ± 338
Number of nuclei in RA cardiomyocytes per mm^2^	2764 ± 416	1803 ± 315 *

Data shown as mean ± S.D. *—significant difference between CONT and MCT rats (*p* < 0.05, Mann–Whitney U test).

## Data Availability

The data presented in this study are available on request from the corresponding author.

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
