# Peer review of "Contractile Behavior of Right Atrial Myocardium of Healthy Rats and Rats with the Experimental Model of Pulmonary Hypertension"

_ijms, 2022, doi:10.3390/ijms23084186_

Round 1
Reviewer 1 Report
The present study mainly demonstrated that atrial slow force response to stretch (SFR) was changed in monocrotaline-induced pulmonary hypertension. Several issues should be clarified. First, the present study used pulmonary hypertension models. Therefore, the title should include "pulmonary hypertension" instead of "heart failure". Second, the authors mentioned in the conclusion that the maladaptive changes in the right ventricular myocardium can be compensated by the elevated atrial SFR. However, there were no data regarding such interpretation.
Author Response
We thank for the reviewing of our manuscript. We carefully addressed all the reviewer's comments in the new version of the manuscript (please see the revised manuscript file where all the changes/modifications are highlighted in red).
Reviewer 2 Report
The manuscript entitled " Contractile behavior of right atrial myocardium of healthy rats and rats with experimental model of heart failure" aim to characterize the contractile behavior and Ca- transient in isolated right atrial strips from healthy rats and rats with pulmonary hypertension in order to reveal functional remodeling of length dependent regulation in failing atrial myocardium. It is an interesting topic and found the alterations in contractility and Ca-transient in the right atrial myocardium of pulmonary pulmonary hypertension rats mostly concern the elevation of slowly developed changes during SFR. However, we suggest use the term " pulmonary hypertension with right heart remodeling" is more suitable than " pulmonary heart disease'.
Author Response

(The authors gave the same response as above.)
